# Design and Clinical Application of an Integrated Microfluidic Device for Circulating Tumor Cells Isolation and Single-Cell Analysis

**DOI:** 10.3390/mi12010049

**Published:** 2021-01-02

**Authors:** Mingxin Xu, Wenwen Liu, Kun Zou, Song Wei, Xinri Zhang, Encheng Li, Qi Wang

**Affiliations:** 1Department of Respiratory Medicine, The Second Hospital Affiliated to Dalian Medical University, No. 467 Zhongshan Road, Dalian 116023, China; xumingxin5426@163.com (M.X.); 18842637303@163.com (S.W.); 2Cancer Translational Medicine Research Center, The Second Hospital Affiliated to Dalian Medical University, No. 467 Zhongshan Road, Dalian 116023, China; liuwenwenphd@163.com; 3Department of Radiation Oncology, The First Hospital Affiliated to Dalian Medical University, No. 222 Zhongshan Road, Dalian 116011, China; zoukun29@163.com; 4Department of Respiratory and Critical Care Medicine, The First Hospital, Shanxi Medical University, No. 85 Jiefang South Road, Taiyuan 030001, China

**Keywords:** CTC-isolation, microfluidics, single-cell analysis, high-throughput sequencing, lung cancer

## Abstract

Circulating tumor cells (CTCs) have been considered as an alternative to tissue biopsy for providing both germline-specific and tumor-derived genetic variations. Single-cell analysis of CTCs enables in-depth investigation of tumor heterogeneity and individualized clinical assessment. However, common CTC enrichment techniques generally have limitations of low throughput and cell damage. Herein, based on micropore-arrayed filtration membrane and microfluidic chip, we established an integrated CTC isolation platform with high-throughput, high-efficiency, and less cell damage. We observed a capture rate of around 85% and a purity of 60.4% by spiking tumor cells (PC-9) into healthy blood samples. Detection of CTCs from lung cancer patients demonstrated a positive detectable rate of 87.5%. Additionally, single CTCs, ctDNA and liver biopsy tissue of a representative advanced lung cancer patient were collected and sequenced, which revealed comprehensive genetic information of CTCs while reflected the differences in genetic profiles between different biological samples. This work provides a promising tool for CTCs isolation and further analysis at single-cell resolution with potential clinical value.

## 1. Introduction

Cancer is a major medical problem endangering human health, while metastasis remains the most common cause of tumor-related death. According to the “seed and soil” hypothesis proposed by Paget in 1889 [1], metastasis is a complex process when rare metastatic “seeds” shed from primary/secondary tumor lesions into the blood circulation and seek suitable “soil” to colonize. These “seeds”, also called circulating tumor cells (CTCs), are considered not only as culprits of metastasis but a promising alternative to tissue biopsy [2]. While enumeration of CTCs has been seen as a prognostic indicator in many cancers, genomic analysis is the prime way to elucidate oncogenic profiles useful for tumor characterization and personalized treatment [3].

In recent years, the rapid development of microfluidics has brought lots of new findings in cellular biology. For cell biomechanics research, the application of microfluidic technology has allowed us to characterize the deformability of live cells, even their nucleuses, to physical stress [4,5]. For drug-related research, with microfluidics, a new approach has been developed and demonstrated for studying the physiological reactions of viable but non-culturable (VBNC) cells to external disturbances, like drug treatment, which may unravel the mechanism of antibiotic resistance [6]. While different procedures have been described to quantify the intracellular accumulation and investigate the membrane-associated transportation of given antibiotics [7]. Similarly, extensive works have been carried out to achieve drug mix and antibiotic diffusion characterization on chip [8,9]. In addition to all the above, microfluidic techniques have also promoted studies of tumor, especially CTCs greatly. Advancements in microfluidic technology have contributed to building numerous new devices for CTCs studies, from detection to cell culture [10]. More importantly, progression in high-throughput sequencing—especially single-cell sequencing—has enabled genomics and transcriptomics analysis of CTCs at single-cell resolution [11]. Single-cell analysis has been broadly used in describing oncogenic mutation patterns, discovering tumor heterogeneities, even guiding treatment options [12,13,14]. Given the emergence of single-cell analysis, it is imperative to establish CTC isolation platforms that are compatible with subsequent genomic analysis. However, common CTC enrichment methods, such as density gradient separation and CellSearch, often have limitations including cell loss, cell damage and time-consuming [15,16]. Therefore, to obtain CTCs qualified for single-cell sequencing, there is an urgent need for a robust device that can achieve high-throughput, high-efficiency CTC isolation while minimizing cell damage.

In this study, we report an integrated device based on micropore-arrayed filtration membrane and microfluidic chip to isolate CTCs for single-cell analysis. The performance of this device has been confirmed on the cultured cell line and patient blood samples of lung cancer. Additionally, to further verify the clinical application of this platform, single-cell sequencing was performed on single CTCs from a representative advanced lung cancer patient.

## 2. Materials and Methods

### 2.1. Microfluidic Chips Fabrication, Filtration Membranes Preparation and System Package

This device consisted of two parts. The micropore-arrayed filtration membranes we used in this study were made of Parylene C by a molding technique. The finished membranes were packaged with Teflon holders, then fixed with magnetic rings before the filtration operation. Details about this part could be found in our previous paper [17].

The magnetic purifying device included a microfluidic chip, a permanent magnet and an electronic control system. According to our previous work [18], the chip was made of polymethyl methacrylate (PMMA) (Sigma-Aldrich, Darmstadt, Germany), and a Laser Engraving Machine (LS100CO2, Gravograph, Shanghai, China) was used for creating microchannels on it. The chip consisted of three layers, which were pressed together for 10 min at 120 °C by a Thermocompressor (CARVER, Muscatine, IA, USA). The magnet was placed under the chip, whose movement could be regulated by an electronic system controlled by a C Language-based software.

### 2.2. CTC Detection

Healthy peripheral blood samples spiked with GFP-expressed PC-9 cells (PC9-GFP) or peripheral blood samples from lung cancer patients (2 mL) were loaded into this device. After filtration, the packaged membrane structure was quickly reversed and 500 μL PBS was added from the upper opening to wash captured blood cells off the membrane, then cells were collected into a tube placed directly below the outlet of the filtration structure along with the fluid. Next, CTCs and residual white blood cells (WBCs) were injected into the purifying chip through the inlet, while CD45 dynabeads were added into it through the outlet. Under the control of the electronic system, dynabeads were forced to run around the reaction chamber of the chip via magnet to bind WBCs, and the best binding time was 20 min. When the reaction was over, the magnet was allowed to stop at the bottom of the reaction chamber to attract the dynabeads, then the supernatant with unbounded WBCs and CTCs was transferred into a culture dish for further identification.

### 2.3. Single-Cell Isolation

A micromanipulator was used for separating single CTCs as described previously [19]. Briefly, for manual cell picking, a tapered ultrathin glass capillary (Friends Honesty Life Sciences Company, Beijing, China) was connected with a Teflon tube to aspirate and dispense single cells, under the negative and positive pressures provided by two constant-pressure pumps (WH-PMPP-12, Wenhao Co., Suzhou, China). The aspirated liquid containing the selected single cells was transferred to a centrifuge tube with lysis buffer for DNA extraction and high-throughput sequencing.

### 2.4. CTCs Identification

The captured CTCs were placed into a cell culture dish with phosphate buffer saline (PBS) for immunofluorescence staining. The CTCs were identified using Alexa Fluor 594 conjugated cytokeratin (Pan-reactive) antibody (1:100, Novus Biologicals, Littleton, CO, USA). Alexa Flour 488 mouse anti-human CD45 (1:100, Abcam, Cambridge, UK) antibody was used to differentiate white blood cells (WBCs). The nuclei were stained with Hoechst (1:2000, Life Technologies, Waltham, MA, USA). Cells were observed under an inverted fluorescence microscope (DMI3000B; Leica Microsystems, Buffalo Grove, IL, USA). CK+/DAPI+/CD45− phenotype was considered to be CTCs, while CD45+/DAPI+/CK− phenotype confirmed WBCs.

### 2.5. Cell Line Culture and Preparation

Human lung adenocarcinoma cell line PC-9, obtained from the American Type Culture Collection (ATCC), was stably transfected with a green fluorescent protein (GFP) before used as a model to mimic actual CTCs in efficacy evaluation of this developed device. Cells were cultured in 10% fetal bovine serum and 1% penicillin/streptomycin added Dulbecco’s Modified Eagle Medium (DMEM)/high-glucose medium (Gibco, Carlsbad, CA, USA; Invitrogen, Carlsbad, CA, USA), with 5% CO_2_ at 37 °C. Cells were incubated with 0.05% trypsin–ethylenediaminetetraacetic acid (EDTA) at 37 °C for 3 min, then suspended and diluted to the desired concentration.

### 2.6. Human Peripheral Blood Samples

The blood samples from lung cancer patients and healthy volunteers were obtained from The Second Affiliated Hospital of Dalian Medical University, People’s Republic of China. All subjects were informed consent. 2 mL peripheral blood of each subject was collected into EDTA containing tubes and processed within 6 h. The clinicopathological characteristics of patients were recorded as well. This study was approved by the Ethics Review Committee of the Second Affiliated Hospital of Dalian Medical University (2018-048). All experiments were conducted according to ethical and safe research practices consisting of human subjects or blood and the Helsinki Declaration of 1975, as revised in 2013.

### 2.7. DNA Sequencing and Bioinformatics Analysis

DNA sequencing was conducted by Haplox (Shenzhen, China). Whole-exome sequencing was performed on liver biopsy tissue, while HapOnco-605 panel sequencing was performed on single CTC and ctDNA samples. The generated library was sequenced on Illumina NovaSeq6000 or HiSeq X platform (Illumina, San Diego, CA, USA), according to PE150 strategies.

The sequencing data was processed by HPS Gene Technology Co., Ltd. (Tianjin, China). FastQC was used to achieve quality control and filter low-quality raw data. The mapping to the human reference genome hg19 was performed with a Burrows-Wheeler Aligner. Somatic SNVs and InDels calling were conducted with muTect and Strelka respectively, while annotation with ANNOVAR. The number of identified SNVs and InDels, together with the proportion of allelic mutations were graphed using R language (ggplot2). R language (maftools) was used to generate the list of genes whose mutation rates ranking top 10. For comparative analysis, R language (VennDiagram) was used to show the number of common or unique mutations/genes among different samples.

### 2.8. Statistical Analysis

All statistical analyses were conducted by GraphPad Prism8. Data were expressed as mean ± standard error of mean (SEM).

## 3. Results

### 3.1. Design and Performance Verification of the Integrated Microfluidic Device

The newly established integrated device was shown in Figure 1. The device consisted of two parts, a filtration system on the top and a magnetic microfluidic chip below. The filtration system included a rotatable packaged micropore-arrayed membrane-based structure for CTC isolation, a removable connecting pipe and a centrifuge tube for the storage of waste or samples remained to be purified. When considered as a whole, this part was about 125 mm high, with the maximum diameter of 40 mm. The position of all components above could be adjusted as needed, and the bottom of the connecting pipe could be placed into the centrifuge tube to prevent liquid splashing. The area of the membrane is 20 × 20 mm, while the diameter of the micropore is 10 μm. As reported in our previous work, the throughput of this membrane could reach up to 17 mL/min for undiluted whole blood, driven by gravity. During filtration, all red blood cells and platelets, as well as most WBCs could pass through the membrane and flow into the waste collection tube. Since the size distributions of CTCs (12–25 μm) and WBCs (5–20 μm) partially overlap [20], CTCs and some WBCs with larger size would be captured on the membrane. The capture rate for lung cancer cells spiked in 5 mL unprocessed whole blood was 83.2 ± 6.2%, with the number of WBCs decreased from 10^9^/mL to 10^4^~10^5^/mL without clogging. After filtration, CTCs and residual WBCs were rinsed off the membrane for further purification. To this end, an automatic magnetic microfluidic chip for negative isolation was employed as described before [18], whose volume was enough to hold 500 μL flushing fluid obtained from the first step. The overall height of the chip and the magnetic base was about 70 mm, and the maximum diameter of the base was 50 mm. The best mixing ratio and binding time for this part had been validated as 10:1 (dynabeads/WBCs) and 20 min in our previous experiments [18]. All parts of this device were fixed on a shelf, whose height and width were 250 and 80 mm respectively.

To test the performance of our integrated device, we spiked PC9-GFP cells into healthy blood samples at concentrations of 10^1^ to 10^4^ cells/mL to mimic actual CTCs (Figure 2a,b). The overall PC9-GFP cells capture rate of this device was calculated as (the number of captured CTCs/the number of total CTCs) × 100%, which was 84.0%, 84.8%, 85.4% and 85.9% for different concentrations (Figure 2c and Appendix A). As for capture purity, which was calculated as (the number of captured CTCs/the number of total isolated nucleated cells) × 100%, we found about 60.4% at the concentration of 10^4^ cells/mL (Appendix A).

### 3.2. Detection of CTCs from Blood Samples of Lung Cancer Patients

For clinical validation of this device, CTC detection was conducted on peripheral blood samples from patients. 16 lung cancer patients of different stages and 4 non-tumor patients as controls, were enrolled in this study. 2 mL blood sample was drawn from each subject, and their clinicopathological characteristics, together with CTC detection results were recorded in Table 1. Captured CTCs were identified by Immunofluorescence staining of CK and CD45. Cells with a phenotype of CK+/CD45−/DAPI+ were recognized as CTCs, while those with CK−/CD45+/DAPI+ were recognized as WBCs (Figure 3a). 14 of 16 lung cancer patients showed positive results of CTC detection (87.5% detectable rate). As shown in Figure 3b,c, the counts of CTCs varied between patients in different stages, while no CTC was observed in any non-tumor patient. 

### 3.3. Single-Cell Analysis of CTC from a Representative Advanced Lung Cancer Patient

To further explore the underlying heterogeneity of lung cancer at single-cell resolution, high-throughput sequencing was performed on circulating tumor DNA (ctDNA), single CTCs and liver metastases from a representative advanced lung cancer patient (patient #1).

As shown in Figure 4, patient #1 was a 42-year-old male who was admitted to the hospital due to cough and blood in the sputum. The findings of computed tomography (CT) imaging and positron emission tomography (PET)-CT both referred to a suspicious malignant nodule in the lower lobe of the right lung, with multiple lesions in the lung, liver and spine. At this time, 34 CTCs were found in his blood sample through our integrated device (2 mL). Several days later, according to the pathological analysis of bronchoscopy and needle biopsies, he was diagnosed with non-small cell lung cancer (NSCLC) with multiple metastases in the liver, bone, and lung (Lung adenocarcinoma, Stage IV). After diagnosis, another CTC detection was performed and 43 CTCs were found in a 1 mL blood sample. Single CTCs were isolated, while ctDNA and the remaining liver biopsy tissue were collected at the same time. Mutation analysis of metastatic liver tissue showed epidermal growth factor receptor (EGFR) mutation, so this patient was given EGFR-tyrosine kinase inhibitor (EGFR-TKI) Tarceva (Erlotinib). After one cycle, significant tumor regression was observed and only 1 CTC was found in a 1 mL blood sample. In the next year, this patient completed 3 sessions of radiotherapy with concurrent administration of Tarceva/Anlotinib and eventually died of sudden breathing difficulties. 4 single CTCs (3 from sample before treatment and 1 from sample after treatment) together with ctDNA and liver biopsy tissue of this patient was sent for DNA sequencing as summarized in Table 2, while his lymphocytes were used as a control to recognize germline-specific genetic variation.

We got sequencing results of 4 samples because the single CTCs before treatment (S090) was contaminated. Through pairwise analysis, we identified 1330 single nucleotide variations (SNVs) and 186 insertion-deletions (InDels) in total. The number of SNVs and InDels of each sample was shown in Table 2, while their distribution on the genome was analyzed in Figure 5a,b. The single CTC sample after treatment (S094) had the highest count of SNVs and InDels, which was mainly distributed in exonic and intronic regions, followed by the tissue sample (S145). As demonstrated in Figure 5c, the base mutation profiles of tissue and single CTC sample were similar, while more complex than that of ctDNA samples, especially the ctDNA sample after treatment (S63). The top 10 somatic mutations among 4 samples were listed in Figure 5d. For tissue/ctDNA samples before treatment (S145 and S124) and single CTC sample after treatment (S094), they all carried CREBBP and EGFR mutation, which were frequently showed in many cancers. EWSR1 and TP53 gene mutation both appeared in two samples respectively, while none of the above 10 genes was found mutated in the ctDNA sample after treatment (S63). Additionally, we evaluated the SNVs, InDels and gene mutations shared between these 4 samples, and the Venn diagram was used to illustrate these results. As shown in Figure 6, there were 4 common mutation sites between tissue and ctDNA samples before treatment (S145 and S124), affecting four known tumor-related genes CREBBP, ROS1, TP53 and EGFR, which might help with the selection of treatment options. Moreover, oncogene HRAS mutated both in single CTC sample and ctDNA sample after treatment (S094 and S063) rather than samples before treatment, suggesting the possible relevance between this gene mutation and targeted therapy.

## 4. Discussion

CTCs are considered to be responsible for the distant metastasis of tumors [21]. As a liquid biopsy, CTC enumeration could predict disease progression and treatment response, while molecular analysis may provide deep insights into mechanisms of tumor metastasis and guide the development of therapeutic strategies [22,23,24]. In recent years, advancements in microfluidic techniques have provided us lots of new tools for cellular biology studies. For example, Pagliara et al. revealed the importance of nuclear structure of embryonic stem cells during their differentiation with microfluidic assay [25], while Cama et al. successfully achieved rapid detection of antibiotic concentration in bacteria by developing a microfluidic platform [26]. Furthermore, the upgradation of materials and refinement of processing have made single-cell studies possible. The combination of microfluidics with imaging equipment have elucidated patterns of long-term growth, division and ageing of Escherichia coli cells [27,28]. While for CTC studies, there have also been many microfluidic devices established to enable more comprehensive understanding of CTCs from a single cell perspective, such as single-CTC metabolic subtypes identification and phenotypes tracking, even targeted proteomics [29,30,31]. However, single-CTCs analysis, especially when combined with high-throughput sequencing, remains a great challenge due to the lack of technologies to obtain intact CTCs qualified for sequencing with high throughput, high efficiency and less cell damage.

In this work, we reported an integrated microfluidic device for CTCs isolation and further single-cell analysis. We observed a capture rate of around 85%, which basically reached the efficiency when using the membrane alone in our previous study [17]. The capture purity of this platform was 60.4% when cell concentration was 10^4^/mL, which was significantly improved than before [17,18]. Then, to test the performance of our device in the clinic, we analyzed blood samples of 16 lung cancer patients and 4 non-tumor patients. We found CTCs in 14 of 16 cancer patients with a positive detectable rate of 87.5%. For one case where we failed to detect CTCs, the medical record revealed that this patient had received surgery and there was no sign of tumor recurrence by the time of detection, which could illustrate the absence of CTCs in his peripheral blood sample. It is worth mentioning that DNA sequencing was performed on single CTCs, ctDNA and liver biopsy tissue of a representative advanced lung cancer patient. Through bioinformatics analysis, we found CTCs hold the most abundant and most complex genetic variations (SNVs and InDels), while ctDNA samples had the least. More importantly, we identified EGFR mutation in liver biopsy tissue and ctDNA sample before treatment, as well as single CTCs after treatment, not in ctDNA sample after treatment. These results suggested that CTCs might be better sources of liquid biopsy than ctDNA, both for tumor heterogeneity research and clinical treatment options. Interestingly, we only found common gene mutations in pre-treatment and post-treatment samples, which indicated the strong impact of treatment on genetic profile. Nevertheless, we cannot draw definitive conclusions relevant to tumor metastasis or treatment response from the current sequencing data, due to the tumor heterogeneity. More studies are needed and the sample size should be expanded to establish the relationship between genetic information and tumor progression. Besides, we are collecting cases similar to patient #1, to get more genetic data from liquid biopsy samples, single CTCs in particular. Findings from this study will help to elucidate the mechanisms of lung cancer liver metastasis.

In this work, we have overcome some limitations addressed in our previous study. First, we replaced the DLD structure with a micropore-arrayed filtration membrane. This change significantly increased CTCs separation throughput of this device without any driving force, such as the pump we used before, while reducing the damage to CTCs caused by fluid shear force and collision with micro-posts. Second, the combination of the filtration system and the CD45 dynabeads based purifying unit dramatically improved the capture purity than using the filtration membrane alone [17], which would decrease the interference of WBCs during single-cell isolation.

However, there remain some difficulties to be addressed. As we can see, the capture rate of this device (around 85%) is slightly lower than using the filtration membrane alone in our previous experiments (83.2–86.7%) [17]. The possible cause of this problem is the cell loss during backwashing, for the cell loss rate of this step in our current work is about 7.3–11.8% (Appendix A). To improve this, we are focusing on surface modification of the filtration membranes with biocompatible materials to achieve the controlled recovery of CTCs. Moreover, the operation of this device requires manual manipulation, which greatly impairs sample processing efficiency. For this issue, we have started to upgrade and automate this device, expecting to achieve translation and clinical application.

To summarize, we have established an integrated platform for CTCs detection and isolation with high throughput and low cell damage. Combining with single-cell methods, our work provides a promising tool for cancer research from a new perspective and paves the way for the implementation of CTC analysis into routine clinical practice.

## Figures and Tables

**Figure 1 micromachines-12-00049-f001:**
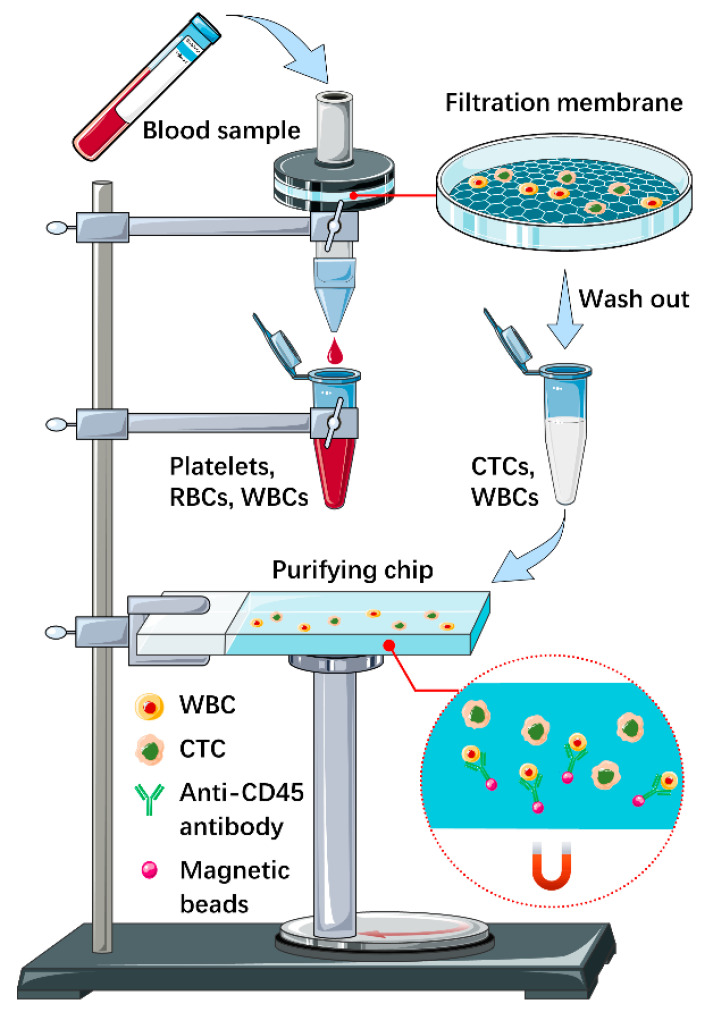
Schematic illustration of the integrated microfluidic circulating tumor cells (CTC) isolation platform. Taking lung cancer as an example, 2 mL blood sample was loaded into the device. All red blood cells (RBCs), platelets and most WBCs were filtered, while CTCs and some WBCs of large size were captured. Then CTCs and residual WBCs were washed off the membrane and transferred into the purifying chip, where WBCs would be further removed by CD45 dynabeads. Finally, the supernatant with CTCs and free WBCs was collected for identification and single-cell isolation. This work is licensed under a Creative Commons Attribution 3.0 Unported License. It is attributed to Mingxin Xu.

**Figure 2 micromachines-12-00049-f002:**
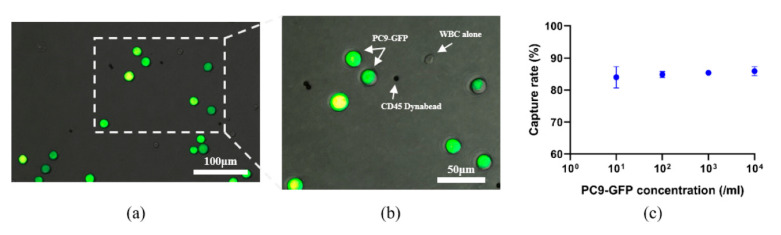
Capture of PC9-GFP cells by the integrated device. (**a**,**b**) Representative merged fluorescent field and bright images of PC9-GFP cells, white blood cells (WBCs) and CD45 dynabeads (as indicated by the arrows) after processing by the novel device (**a**) (magnification, ×200), (**b**) (magnification, ×400). (**c**) Capture rate of the integrated device at different concentrations of PC9-GFP cells. Data were expressed as mean ± standard error of mean (SEM).

**Figure 3 micromachines-12-00049-f003:**
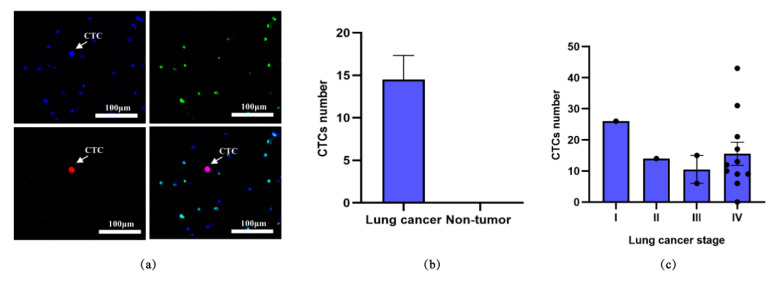
Analysis of captured CTCs from blood samples of patients. (**a**) Representative fluorescent staining images of CTC detected in a blood sample of a lung cancer patient. The blue color referred to nuclear staining (DAPI), the green color referred to CD45, and the red color demonstrated CK expression. Magnification, ×200. (**b**) The average number of CTCs in lung cancer patients (*n* = 16) and non-tumor patients (*n* = 4). (**c**) The average number of CTCs in lung cancer patients of different stages (Stage I, *n* = 1; Stage II, *n* = 1; Stage III, *n* = 2; Stage IV, *n* = 11).

**Figure 4 micromachines-12-00049-f004:**
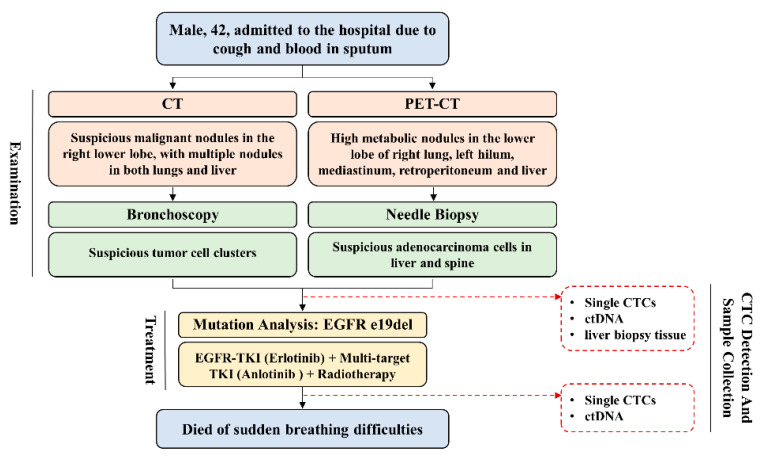
Schematic illustration of the diagnosis, treatment and sample collection of a representative advanced lung cancer patient (patient #1).

**Figure 5 micromachines-12-00049-f005:**
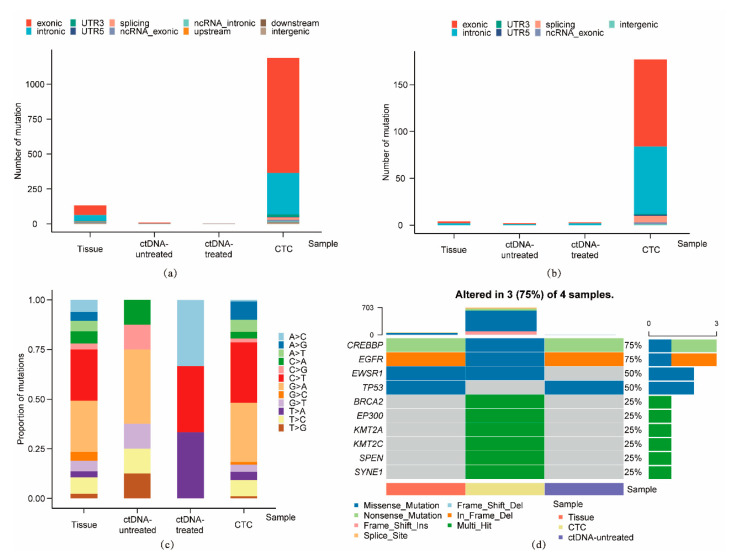
The genetic variations of liver biopsy tissue, ctDNA and single CTC samples. (**a**) The distribution of SNVs on the genome. (**b**) The distribution of InDels on the genome. (**c**) The proportion of different base mutation types. (**d**) The top 10 somatic mutations among 4 samples.

**Figure 6 micromachines-12-00049-f006:**
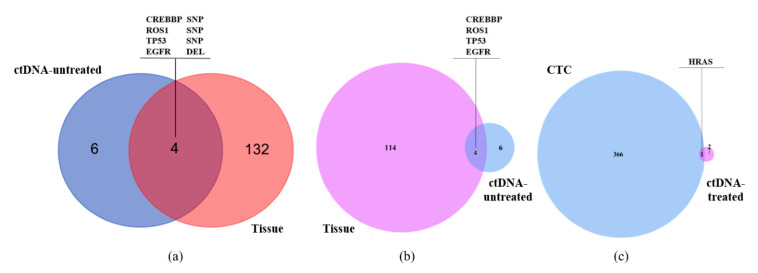
Common SNVs, InDels and mutated genes in liver biopsy tissue, ctDNA and single CTC samples. (**a**,**b**) Common mutation sites and related genes in ctDNA and liver biopsy tissue samples before treatment (S124 and S145). (**c**) Common gene mutation in single CTC and ctDNA sample after treatment (S094 and S063). SNP: single nucleotide polymorphism; DEL: deletion.

**Table 1 micromachines-12-00049-t001:** Clinical characteristics of 20 patients.

No.	Gender	Diagnosis	Stage	Histological Type	CTC Count (N)
1	Male	Lung cancer	IV	Adenocarcinoma	43
2	Female	Lung cancer	IV	Adenocarcinoma	21
3	Male	Lung cancer	IV	Small cell carcinoma	6
4	Male	Lung cancer	IV	Large cell carcinoma	10
5	Female	Lung cancer	IV	Adenocarcinoma	31
6	Male	Lung cancer	IV	Adenocarcinoma	9
7	Male	Lung cancer	IV	Adenocarcinoma	0
8	Female	Lung cancer	IIIC	Adenocarcinoma	15
9	Female	Lung cancer	IV	Adenocarcinoma	13
10	Male	Lung cancer	Postoperative	Adenocarcinoma	0
11	Male	Lung cancer	IIA	Adenocarcinoma	14
12	Female	Lung cancer	IV	Adenocarcinoma	9
13	Female	Lung cancer	IV	Adenocarcinoma	12
14	Male	Lung cancer	IV	Squamous cell carcinoma	17
15	Female	Lung cancer	IIIB	Adenocarcinoma	6
16	Male	Lung cancer	IA	Adenocarcinoma	26
17	Male	Pneumonia	— ^2^	— ^2^	0
18	Male	COPD ^1^	— ^2^	— ^2^	0
19	Male	COPD ^1^	— ^2^	— ^2^	0
20	Male	COPD ^1^	— ^2^	— ^2^	0

^1^ COPD, chronic obstructive pulmonary disease; ^2^ “—”, no such information.

**Table 2 micromachines-12-00049-t002:** Sequencing information of samples from patient #1.

Sample ID	Sample Type	Sequencing Technique	Number of SNVs	Number of InDels
S145	Liver biopsy tissue	Whole exome sequencing	132	4
S124	ctDNA before treatment	HapOnco-605 gene panel	8	2
S090	3 single CTCs before treatment	HapOnco-605 gene panel	–	–
S063	ctDNA after treatment	HapOnco-605 gene panel	3	3
S094	1 single CTC after treatment	HapOnco-605 gene panel	1187	177

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
