# Peer review of "Design and Clinical Application of an Integrated Microfluidic Device for Circulating Tumor Cells Isolation and Single-Cell Analysis"

_micromachines, 2021, doi:10.3390/mi12010049_

Round 1
Reviewer 1 Report
This study designed an integrated system for CTC isolation from both spiked and clinical samples and provided single-cell sequencing. The writing is easy to follow, but the paper can be improved by addressing my following comments:
Major comments:
- The authors need to clearly define the parameters they used to assess CTC isolation in line 159-163, page 4, including capture rate, clearance efficiency, capture purity and recovery rate. The authors need to describe how these parameters are obtained and calculated as well as providing the raw data/number (maybe in a table in Supporting information). Also, is the capture rate and recovery rate the same thing or not?
- The CTC count in Table 1 need to be presented in a figure to better compare cancer and non-cancer patients and if there is any trend between different cancer stages.
- In line 210, page 7, the authors mentioned that the data for single CTCs before treatment was contaminated. Is it possible to obtain this data again? As the emphasis of this paper is about CTCs instead of tissue/ctDNAs, this data is especially important to compare before and after treatment.
- How many CTCs were analysed in the sequencing to get the data? The importance of single CTC analysis is to compare the heterogeneity between individual cells, which can be obtained by sequencing different individual cells. However, this important data was not provided here.
- In figure 4, it is hard to compare the results by using sample ID as the x-axis. It is better to compare if using abbreviations with meanings that can be easily identified.
- Figure 1 should be modified so that the purifying chip is right under the tube of CTC (right one), not the tube of RBC (left one). Also, use colors with more contrast (red and green) for WBC and CTC, right now the blue and green are hard to tell.
Minor comments:
- Line 34-35, page 1: Replace ‘tumor’ with ‘cancer’.
- Line 60, page 2: unnecessary capital letters for subtitle 2.1
- Line 80, page 2: add the duration of the binding
- Line 150, page 4: add reference
- Figure 2 caption in page 5: add the meaning of the error bar
- Table 2 in page 6: add CTC numbers. Combine table 2 and 3 in one table
- All figures: the panel letters (a,b, c…) should appear at the top left corner of each figure
- Line 261 in page 9: should ‘except for’ be replaced with ‘not in’?
- Line 279 in page 9: add the number of the capture rate of this device and in previous work.
Reviewer 2 Report
In this manuscript, the authors developed a CTC isolation platform using micropore-arrayed filtration membranes and spiking PC9-GFP cells into healthy blood 159 samples at concentrations of 10 to 10,000 cells/ml to mimic actual CTCs. My specific concerns are as follows.
- Where does the filtration membranes purchase? Please address the property of the filtration membranes even though the authors referred it to reference [11]. It is essential to be addressed because of the sample preparation.
- How to prevent the micropore-arrayed filtration membranes from clogging? Please address the typical sizes for RBCs, WBCs, PC9-GFP cells, as well as the size of practical CTCs.
- It lacks a convincing demonstration by using the real patient’s blood sample. Mainly I did not recognize the novelty of the chip design. The real patient’s blood sample might be necessary for this paper.
- The magnetic sorting of CTCs has been reported on a few papers and is also commercially available (FDA approved). I do not recognize the novelty of these reported results.
- In line 142, it mentions that CTCs and some larger WBCs would be captured on the membrane, and in line 162, above 99.9% WBCs will be removed. However, the capture purity is 60.4%. What is the other 39.6%?
- What is the possible CTC concentration of the blood sample for blood samples testing from lung cancer patients?
- In table 3, what is the meaning of total SNVs and total InDels? Also, some of them are the same. Why are they count twice?
- What is the meaning of figure 4-C? The cells are different. The numbers of mutations are also different. Nucleotide mutations do not affect the same genes.
Reviewer 3 Report
This manuscript introduces a novel combined filtration and microfluidic device for characterising the cellular heterogeneity in circulating tumours. I believe that the manuscript will be of interest to the readership of Micromachines, however the following issues must be addressed:
1) The introduction on microfluidics is rather short and limited in scope. The authors should expanded by reporting other recent findings in cellular biology using microfluidics. For example, in terms of heterogeneity in:
- Cell biomechanics:
Proceedings of the National Academy of Sciences 109, 7630 (2012)
Nature materials 13, 638 (2014)
Lab on a chip 17, 805 (2017)
- Drug resistance:
BMC Biology, 15, 121 (2017)
Science 356, 311 (2017)
- Membrane transport:
Nature protocols 13, 1348 (2018)
Lab on a chip 20, 2765 (2020)
- Cellular ageing:
Proceedings of the National Academy of Sciences 105, 3076 (2008)
Philosophical Transactions of the Royal Society B 374, 20180442 (2019)
- Similarly, extensive work has been carried out on antibiotic diffusion and mixing on chip but this manuscript lacks an in depth study of the literature including:
Lab on a Chip, 5, 974 (2005)
Lab on a Chip 14, 2303 (2014)
- Lines 45-46: there are several other methods besides ref 12-13 allowing confinement and tracking of individual bacteria, for example, the following papers should be cited:
Curr Biol 20: 1099 (2010)
Phil. Trans. R. Soc. B 374: 20180442 (2019)
2) Section 3.1. The authors should state the overall dimensions of their combined filtration and microfluidic device. Is this portable? The filtration throughput should be converted to number of cells per minute
3) Figure 3 is of very poor resolution especially 3b, this should be addressed
4) Same issue in Figure 4
5) The discussion should be broaden in scope taking into account the cellular heterogeneity reported ion my point 1 above
Round 2
Reviewer 1 Report
Comments are addressed well and can be accepted.
Reviewer 2 Report
This revised version is ok to me.